# Camera Calibration Using Gray Code

**DOI:** 10.3390/s19020246

**Published:** 2019-01-10

**Authors:** Seppe Sels, Bart Ribbens, Steve Vanlanduit, Rudi Penne

**Affiliations:** Faculty of Applied Engineering Department Electromechanics, Universiteit Antwerpen, Groenenborgerlaan 171, 2020 Antwerpen, Belgium; Bart.Ribbens@uantwerpen.be (B.R.); Steve.Vanlanduit@uantwerpen.be (S.V.); Rudi.Penne@uantwerpen.be (R.P.)

**Keywords:** camera calibration, Gray code, checkerboard

## Abstract

In order to determine camera parameters, a calibration procedure involving the camera recordings of a checkerboard is usually performed. In this paper, we propose an alternative approach that uses Gray-code patterns displayed on an LCD screen. Gray-code patterns allow us to decode 3D location information of points of the LCD screen at every pixel in the camera image. This is in contrast to checkerboard patterns where the number of corresponding locations is limited to the number of checkerboard corners. We show that, for the case of a UEye CMOS camera, the precision of focal-length estimation is 1.5 times more precise than when using a standard calibration with a checkerboard pattern.

## 1. Problem Statement and Introduction

Commonly, camera calibration is done with checkerboard patterns printed on paper with a standard printer and attached to a flat surface [1,2]. The use of these low-cost checkerboards limits the number of input points of the calibration to the number of checker corners. It is also challenging to obtain input points of pixels located near the edges and corners of the image. This difficulty might lead to high uncertainty on the calculated camera parameters. The primary goal in this work is to obtain more accurate camera calibration by using a new calibration pattern. The proposed pattern needs to give a high number of accurate input points to the calibration algorithm. The input points must be easy to detect and evenly distributed in the image frame to avoid overfitting. In addition, the used calibration pattern must be easy to handle, cheap, and easy to make. Therefore, a standard laptop LCD screen (or any other screen) with a displayed pattern is proposed. A laptop screen is a high-quality flat surface and easy to get by. As a pattern, we propose to use Gray code. In our experiments (Section 2) a Gray-code pattern performs better than a checkerboard pattern. The main advantage of Gray code is that each pixel of the image has a corresponding calibration input point. The use, advantages, and disadvantages of Gray code are further explained in Section 1.3. In Section 1.1, a brief introduction on the standard camera-calibration method is given. Section 2 describes the experiments used to validate the proposed calibration board. Section 2 also compares the proposed Gray-code pattern with a checkerboard pattern. The literature exists where patterns displayed on LCD screens are used [3,4,5], but they are mostly used in combination with checkerboards and circular patterns, or need additional hardware. To our knowledge, Gray code is not used as a calibration pattern in combination with standard pinhole monocular camera calibration. The closest related work is the paper by Hirooka S. [6]. The work uses Gray code displayed on LCD screens to calibrate a stereo pair of cameras. In contrast to the paper of Hirooka S. [6], this work focuses on monocular calibration. Additionally, we analyze the precision of the calibration and compare it to standard checkerboards displayed on an LCD screen. The standard error measure in camera calibration (reprojection error) of the standard method using checkerboards and the proposed method using Gray code is investigated.

### 1.1. Camera Calibration

Geometric monocular camera calibration plays an important role in computer vision [7,8,9,10,11,12,13,14]. This work focuses on calculating Intrinsic Parameters I and distortion parameters of the camera. These parameters are needed if images are used for pose estimation of detected objects or 3D reconstruction (scanning) of objects *K* is the intrinsic matrix according to the pinhole model assuming compensated distortion and zero skew (Equation (Equation 1)). The intrinsic matrix contains the focal lengths [pixel] in the x- and y-direction ($f_{x},f_{y}$) and the principal point ($c_{x},c_{y}$) [pixel]. The focal length in pixels corresponds to the focal length in meter with $f_{x,y}=s_{x,y}*f$. Where *f* is the focal length (meters), and $s_{x,y}$ (pixels/meters) is the size in x or y direction of a pixel on the camera sensor. The goal of camera calibration is to calculate this intrinsic matrix and the distortion parameters. As input points the calibration algorithm uses known world co-ordinates and corresponding camera coordinates. Commonly, these co-ordinates are called imagepoints(u,v) (Equation (Equation 2)) when used as calibration points. $Objectpoints\;{\left(x,y,z\right)}_{world}$ (Equation (Equation 3)) are points corresponding with these imagepoints defined in a world co-ordinate system.

(1)$$K=\left[\begin{array}{ccc} f_{x} & 0 & c_{x}\\ 0 & f_{y} & c_{y}\\ 0 & 0 & 1 \end{array}\right]$$

(2)$$\left[\begin{array}{c} u\\ v\\ 1 \end{array}\right]=K\left[\begin{array}{c} x\\ y\\ z \end{array}\right]$$

*H* is the homogeneous transformation matrix between the objectpoints defined in a world co-ordinate system and the corresponding points in the camera co-ordinate system. For more information about the pinhole model with (radial) distortion parameters, we refer to the work of Zhang Z., OpenCV and Matlab documentation [1,15,16].

(3)$$\left[\begin{array}{c} x\\ y\\ z\\ 1 \end{array}\right]=H.{\left[\begin{array}{c} x\\ y\\ z\\ 1 \end{array}\right]}_{world}$$

Currently, object and imagepoints for camera calibration are commonly obtained using planar checkerboards [11,16,17]. From the images of these boards, the checker corners are calculated and located. This location is used to estimate camera parameters, where the camera parameters contain the intrinsic matrix and distortion parameters. There are various kinds of checkerboards. The most common is a board used in OpenCV tutorials (see Figure 1a). Other types of boards exist (Figure 1b–d) using circles instead of corners. Boards with additional markers to detect partially occluded checkerboards also exist (Figure 1c,d). These types of calibration boards all have a limited number of detectable points per image, e.g., the 9 × 6 OpenCV checkerboard has 54 detectable points [11], where even a low-resolution camera has 76,800 pixels (320 × 240 pixels). Using a limited set of points can cause inaccurate calibrations with a large uncertainty (see Section 2).

### 1.2. LCD Screen

Commonly planar checkerboards or other patterns are made by printing the pattern on paper using a standard printer and then placing or glueing the paper on a planar surface [15,16]. This manufacturing process can introduce errors caused by the printing or glueing process [2]. To avoid problems with this manufacturing process, the pattern can be displayed on an LCD screen [3,19]. An LCD screen is flat, and the displayed dimensions of the pattern are highly accurate and without distortions. Song Z. [5] states that the flatness of a printed checkerboard easily exceeds 0.1 mm, while the planetary deviations of standard LCD panels are below 0.05 mm. High-quality machine-vision calibration targets exist (using ceramic or glass substrates), but their price range is much higher than calibration boards printed on paper and manually glued on a flat surface. However, in this work, it is not our goal to replace these industrial-grade calibration boards, but only the low-cost printed calibration boards.

### 1.3. Gray Code

In this work, we replace a standard checker calibration pattern with a pattern that gives more correspondences for calibration. As a new pattern, we propose to use Gray code [20] (reflected binary code). Commonly, Gray code is used for error detection and correction in digital communication. It is also used in encoders for position detection and used in structured light 3D scanners [21,22,23,24,25]. In structured light 3D scanners, Gray code is a type of binary code that uses black and white stripes. Each stripe corresponds to a unique code word. *N* patterns can code $2^{N}$ unique patterns. The sequence of these striped patterns codes $2^{n}$ unique locations. In the proposed methodology, each pixel of the camera is mapped to a row/column of the LCD screen. To calculate this mapping, Gray code is displayed on a screen and captured by the camera. Gray code encodes column and row indices in a unique time sequence of black and white patterns. This pattern forms code words (see Figure 2). Each pixel of the camera will have two codewords, one corresponding with the row of the LCD screen it sees, and one with the column. Two neighboring code words have a Hamming distance of one. This property has the advantage that if one frame of the Gray-code sequence is detected wrongly, the corresponding pixel only shifts one row/column [20,26]. In contrast to standard checkerboard detection, multiple frames are recorded to calculate the mapping. This recording has as a disadvantage that the camera needs to remain fixed during recording. The number of displayed frames is equal to $\left[2\log_{2}\left(w\right)+2\log_{2}\left(h\right)\right]$ where *w* is the width of the LCD screen and *h* the height (in pixel). The displayed frames are the bit planes needed for decoding and their inverse. This inverse is used to define a variable threshold to distinguish black (0) and white (1) [26]. Checkerboard detection requires the detection of corners on a subpixel level. Standard techniques use a Harris feature point detector as a rough corner estimation and use a gradient search operation for subpixel corner estimation. As proven by Datta A. [27], this gradient search introduces a bias on the corner location when the calibration boards are not frontoparallel to the camera. Although the work of Datta A. solves this by using an iterative approach, the use of Gray-code patterns eliminates this bias because only a threshold operation is needed for composing the per pixel Gray-code code word. This per-pixel threshold does not use line detection or other spatial information about the calibration pattern that might get distorted in the image. As a downside, Gray code only gives only pixel correspondences in contrast to subpixel correspondences of checkerboard detection. There are, however, a lot more pixel correspondences (all pixels of the image can be used) than the 54 points used by a standard 9 × 4 checkerboard. The camera can be placed in a position where it only sees the LCD screen because it is not necessary to see the complete pattern. This property gives the advantage that each camera pixel has a correspondence that also forces the correspondences to be evenly distributed.

When using Gray code displayed on an LCD screen, the camera needs to operate in the visible light spectrum, where checkerboards can also be used to calibrate camera and lenses in the infrared and near-infrared spectrum [28]. When recording a screen with a camera, special care needs to be taking to avoid incompatibility with refresh rates of the LCD screen and capturing rate of the camera. In our experiments, aliasing effects like the moiré effect did not occur. The methodology is summarised in Algorithm 1. An example of the first three Gray-code patterns is given in Figure 2.   

**Algorithm 1:** Methodology.
Step 1:Aim camera to LCD screen.Step 2:Display and Capture Gray code pattern.Step 3:Decode Gray code.
Step 3.a:Make mapping between screen pixels and camera pixels.
Step 4:Redo 1–3 for multiple positions.Step 5:Use OpenCV functions for camera calibration.
Step 5.a:Sample complete dataset (otherwise large computation time).Step 5.b:Use OpenCV function calibrateCamera (inputs) with as input imagepoints (u,v coordinates of image) and objectpoints (corresponding screen pixels).



## 2. Experiments

### 2.1. Experimental Setup

In our experimental setup, a UEye CMOS camera with a resolution of 1024 × 1080 pixels was used in combination with a 16 mm lens. The camera sensor has a pixel size of 5.3 µm, which gives a theoretical focal length of 3018 pixels. As LCD screen, the screen of a Dell Lattitude E5550 (1080 × 1920 pixels) was used. In the experiment, the method using Gray code is compared with checkerboards.

Checkerboard detection uses additional circle markers on the board (Figure 3b). These markers are used as a reference to ensure correct labelling (ordering) of the corners when the complete checkerboard is not visible (Figure 3d). In our experiments, we use a 9 × 14 checkerboard. A checkerboard detection is considered correct when minimal 54 points are detected (total points 126). 54 points are used as a threshold to do equally or better than the standard calibration board used in OpenCV tutorials (9 × 6, 54 points) [1].

For each calibration, the camera is repositioned 15 times. We chose this number because OpenCV documentation (version 3.4.1) states that at least 10 positions are needed. Matlab documentation (using the same algorithms) states that 10 to 20 positions are needed. Calibration software like DLR [29] only needs three to 10 images. Halcon (industrial vision software package) also uses 15 positions of a calibration board in its documentation [30]. To obtain proper calibrations, we positioned the camera with angles up to 30° as done by Albarelli A. et al. [2].

Display and decoding of the Gray code is done using the Matlab Psychtoolbox (Matlab OpenGL wrapper) and code made available by Moreno D. [31] (Figure 3a,c). Calculating the calibration is done using OpenCV (version 3.4.1) functions. The function uses the calibration algorithm of Zhang Z. [15]), and default settings are used. Although not displayed in the text, radial distortion (to the 2nd order) and principal points are calculated. Skew and tangential distortion, and third-order radial distortion are assumed to be negligible.

### 2.2. Number of Calibration Points

In a first experiment, the effect of a higher number of calibration points was analysed. From a dataset of 15 camera positions, N random number of samples per camera position are taken, and the calibration is executed. In Figure 4 different calibration results (focal length) with a different number of samples are listed. For each sample size, the calibration is repeated 50 times. The samples are randomly selected from a complete dataset containing all samples from 15 different positions of the camera. The experiment shows that, when using more than 100,000 points, standard deviation goes below 0.14 pixels, and the mean focal length does not change. When using 700,000 points or more, standard deviation stays below 0.02 pixels. Therefore, in the following experiments, 700,000 points are used as input sample size. Calculating the calibration with 700,000 points takes approximately two minutes (computation done on 1 core @ 4 GHz). In comparison, a calibration using 54 points per view (checkerboard) takes less than one second.

### 2.3. Comparison between Gray Code and Checkerboard

In this experiment, in 15 different camera positions, Gray code was recorded and subsequently decoded. In each camera position, a checkerboard is also displayed on the screen. Next, the camera was calibrated using both the detected checkerboard corners and a random subset (700,000 points) of the Gray-code pattern. Next, the experiment was repeated seven times to check repeatability.

Table 1 shows the results of the experiment. The reprojection error of the Gray code was higher than the reprojection error calculated with checkerboards. This higher error is expected since the checkerboard detection has subpixel resolution, whereas Gray code only has pixel correspondences. The reprojection error is the root mean square of the distances between the detected *imagepoints* and the projected (using the calculated camera parameters) *objectpoints* (see Section 1.1). The experiments of Albarelli A. et al. [2] and Poulin-Girard A. et al. [32] also prove that low reprojection errors do not necessarily give better calibration. The positioning of the calibration boards proved to be far more important to get good calibrations. For example, large calibration errors with low reprojection errors are obtained using only positions planar to the calibration. Therefore, we used standard deviation on the calibration results to compare results.

In Table 1, the comparison between the use of a checkerboard and Gray code is summarised. From the complete set, the mean focal length (in x-direction) calculated with Gray code is 3091 pixels with a standard deviation of 2.3 pixels, while the calculated focal length using a checkerboard pattern is 3080 pixel with a standard deviation of 4.8 pixels. In this dataset, the use of Gray code gives an improvement of 2.5 pixels on standard deviation.

As an extra validation, an LCD screen displaying a checkerboard is mounted on a translation stage. The checkerboard is detected in 20 different positions between 200 mm and 500 mm from the camera. Between each pair of positions, there was a translation of 10 mm. Next, the relative translation of the checkerboard is calculated with the mean calibration parameters of Table 1 of both the method using Gray code and the method using checkerboards. The mean relative error on the translation with the Gray-code parameters was 0.78 mm, with a standard deviation of 0.32 mm. The mean relative error using checkerboard parameters was 1.65 mm, with a standard deviation of 0.93 mm.

The experiment shows that a large number of correspondences has the advantage to give smaller standard deviations on the estimated camera parameters. Where the checkerboard gives standard deviations on the focal length of around 2.0 pixels, the Gray code gives standard deviations of around 0.02 pixels which is an improvement of a factor 100. Therefore, we conclude that using Gray code makes the camera-calibration algorithm more robust than calibration using correspondences from checkerboards. Note that the iterative method of Datta A. [27] (see Section 1.3) has an approximate improvement of a factor 3.

### 2.4. Repeatability

To further check the repeatability of the calibration, stability analysis was executed. A new calibration set was built with 49 different camerapositions. From this set, seven random camera positions are chosen. In this subset, 700,000 points were randomly selected, and then calibration was calculated. In the case of the checkerboard, the same camera positions are used to calibrate the camera with all detected checkerboard corners. This analysis was repeated 100 times and the focal length ($f_{x}$) is saved.

In this analysis (see Figure 5), the standard deviation of the focal length using Gray code was 3.30 pixels, which is lower than the focal length calculated with checkerboard corners (5.87 pixels). Note that in this experiment samples are taken randomly. Consequently, it is possible to build a bad dataset with only low-angled calibration patterns. For that reason, this experiment would most probably give higher deviations than real experiments done by an experienced user. However, since the same sampling is used for both techniques, the experiment can be used as a comparison. Note that in this dataset the lens of the camera was refocused and consequently the lens would have a slightly different focal distance compared to previous experiments.

The used Gray-code pattern uses 44 images (22 for rows, 22 for columns) displayed on the screen per position. As an additional experiment, 44 different checkerboards were also detected per camera position. This experiment did not significantly alter the calibration results with checkerboards and calculating the calibration with this dataset takes between 350 and 60 min. Due to this high calculation time, and the fact that this is not a standard use of checkerboard patterns, this setup of 44 checkerboards per camera position was not further analysed. For all other experiments described in this work, only one checkerboard was detected per camera position.

## 3. Conclusions

A Gray-code pattern displayed on a standard LCD screen can be used for the geometric calibration of a camera. In our experiments, the average reprojection error was around 1 pixel. This error is larger than the error obtained using standard checkerboards (0.25 pixels). This higher error is caused by the pixel-accuracy detection of Gray code, whereas the checkerboard detection uses subpixel detection of the corners. The Gray-code pattern has the advantage of giving a large number of correspondences. This large number made it possible to give each camera pixel a world co-ordinate in our experiments. This is in contrast to the correspondences of a checkerboard that are limited to the checkerboard corners. A large number of correspondences has the advantage to give smaller standard deviations on the estimated camera parameters. Where the checkerboard gives standard deviations on the focal length of around 4.8 pixels, the Gray code gives standard deviations of around 3.1 pixels, which is an improvement of a factor of 1.5. Therefore, we conclude that using Gray code makes the camera-calibration algorithm more robust than calibration using correspondences from checkerboards. Calibration does not need special lab-equipment, only a tripod for mounting the camera and a laptop or LCD monitor.

## Figures and Tables

**Figure 1 sensors-19-00246-f001:**
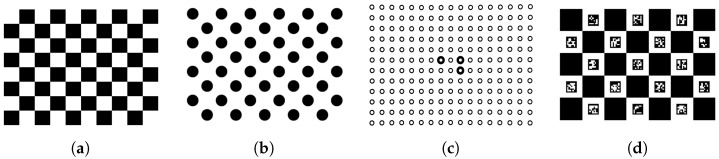
Calibration Boards. (**a**) Opencv 9 × 6 checkerboard; (**b**) Opencv asymmetric circle calibration board; (**c**) board with additional (bold circles) markers (used by Scan in box Idea software); (**d**) Charuco markers in combination with Checkerboard (OpenCV) [1,18] (edited figure from OpenCV documentation).

**Figure 2 sensors-19-00246-f002:**
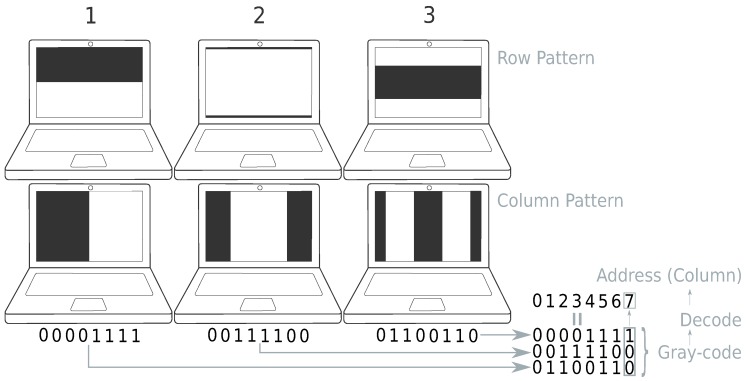
Gray-code pattern (first 3) Top: Pattern for row correspondences (top), column for row correspondences. An example of decodation of the column pattern is given. The example assumes only three patterns are given (eight columns). Figure based on Reference [6].

**Figure 3 sensors-19-00246-f003:**
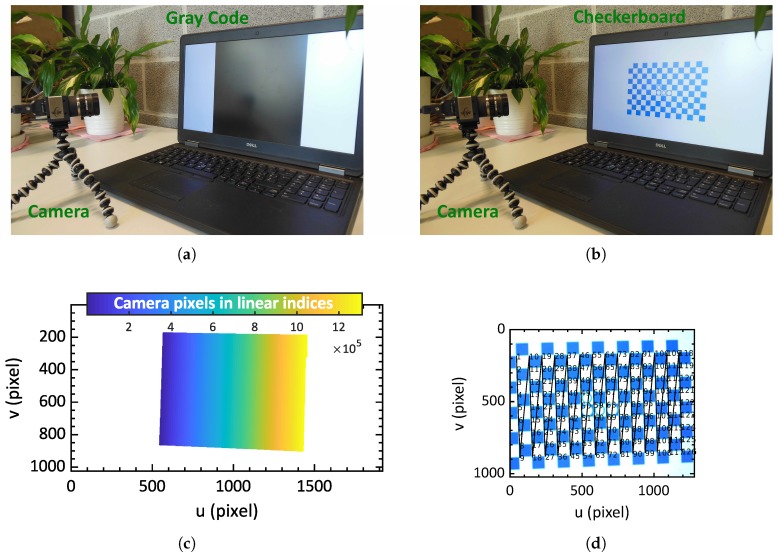
(**a**) Setup with one Gray-code frame displayed (**b**) setup with checkerboard displayed (**c**) recorded frame with Gray code; (**d**) calculated correspondence map between LCD screen and camera. Each pixel on the LCD screen (x and y-axis) has a corresponding pixel in the camera. The corresponding pixels are visualized with colors. The white area indicates the parts of the screen that is not visible in the camera image.

**Figure 4 sensors-19-00246-f004:**
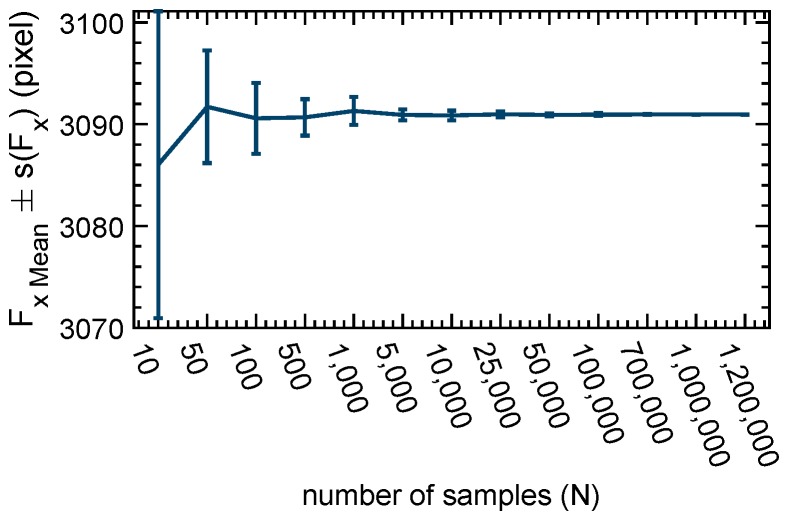
Calibration using Gray code with a different number of inputpoints.

**Figure 5 sensors-19-00246-f005:**
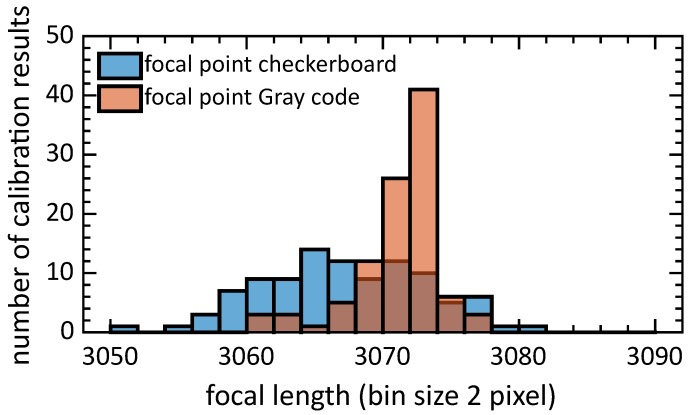
Histogram of focal length (($f_{x}$) x-direction) using a calibration by sampling seven random positions out of 49, executed 100 times. Standard deviation using Gray code is 3.30 and 5.87 pixels using standard checkerboard displayed on a LCD screen

**Table 1 sensors-19-00246-t001:** Experimental results of seven independent calibrations. Each calibration is calculated from 15 (new) positions. $f_{x}$ is the focal length in the x-direction, $f_{y}$ is the focal length in the y-direction. r1,r2 are the distortion parameters (first and second order radial distortion). $c_{x}$ and $c_{y}$ are the principal points. With individual calibrations $s,(parameter_{*})$ is the standard deviation estimated from the intrinsic parameters. $s(parameter_{*})$ of the mean value is the standard deviation calculated with the respective parameter of each calibration.

Type	$f_{x}$	*s*($f_{x}$)	$f_{y}$	*s*($f_{y}$)	Mean Reprojection Error	*r*1	*s*(*r*1)	*r*2	*s*(*r*2)	$c_{x}$	*s*($c_{x}$)	$c_{y}$	*s*($c_{y}$)
	[pixel]	[pixel]	[pixel]	[pixel]	[pixel]					[pixel]	[pixel]	[pixel]	[pixel]
Gray-code	3093.1	0.10	3091.7	0.09	1.15	−0.14	0.26	0.21	0.0006	659.80	0.04	511.27	0.07
	3095.0	0.11	3094.7	0.10	1.01	−0.16	0.44	−0.92	0.0006	654.93	0.04	509.43	0.08
	3087.9	0.07	3087.7	0.06	0.88	−0.16	0.46	−1.12	0.0005	655.62	0.03	512.87	0.05
	3091.9	0.08	3091.2	0.07	0.93	−0.13	0.02	2.28	0.0005	658.27	0.03	513.02	0.06
	3090.9	0.10	3089.9	0.09	0.80	−0.14	0.20	0.87	0.0005	657.81	0.04	510.73	0.06
	3085.7	0.07	3091.7	0.09	1.15	−0.14	0.26	0.21	0.0006	659.80	0.04	511.27	0.07
	3091.0	0.07	3090.9	0.06	0.94	−0.15	0.44	−0.76	0.0005	653.75	0.03	514.89	0.05
mean ± s	3090.8 ± 3.1		3091.1 ± 2.1		0.98 ± 0.13	−0.15 ± 0.01		0.11 ± 1.2		657.14 ± 2.4		511.93 ± 1.8	
checkerboard	3080.1	2.13	3079.8	1.87	0.17	−0.16	0.85	−7.73	0.0097	658.80	0.78	510.53	1.74
	3079.8	2.87	3080.0	2.70	0.37	−0.16	0.61	−5.29	0.0188	658.52	1.59	513.45	2.42
	3088.7	1.74	3087.7	1.52	0.20	−0.17	1.02	−7.99	0.0125	659.64	0.85	508.38	1.43
	3078.4	2.07	3078.7	1.87	0.21	−0.15	0.69	−6.76	0.0113	656.52	0.93	519.11	1.38
	3082.0	3.87	3081.9	3.53	0.23	−0.16	1.31	−14.33	0.0165	656.99	1.21	515.91	2.53
	3073.0	7.70	3074.4	7.44	0.39	−0.17	1.28	−12.97	0.0293	656.24	2.83	524.64	4.49
	3077.3	2.22	3078.0	2.06	0.21	−0.16	0.90	−6.73	0.0120	660.55	1.03	519.41	1.46
mean ± s	3079.9 ± 4.8		3080.1 ± 4.1		0.25 ± 0.09	−0.16 ± 0.01		−8.83 ± 3.43		658.18 ±1.64		515.92 ± 5.63	

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
