# Peer review of "Camera Calibration Using Gray Code"

_sensors, 2019, doi:10.3390/s19020246_

Round 1

Reviewer 1 Report

Authors propose approach to camera calibration by using active LCD screen from laptop. Presented approach is to naïve in present form. I have several mandatory points that needs to be addressed before publication:

1)      Zhang calibration should be performed with about 20 images – not seven, why authors used 7?

2)      How did you get the reference focal length in pixels?

3)      Please provide full camera calibration followed by measurement of real reference distances. Only by measuring of known distance you can prove that your calibration is better.

4)      Please discus and provide reference data about flatness of your calibration model. According to my knowledge flatness of calibration model and accuracy of marker detection are main factors of calibration quality. In practice for precise measurement the special calibration bodies are manufactured and measured by reference method. Later knowing their exact dimension calibration is performed. How you measured planarity error of your calibration model?

In abstract you have “whick” word.

Author Response

Thank you for your interest and reviewing the paper. Below you can find answers to your comments and suggestions.

Authors propose approach to camera calibration by using active LCD screen from laptop. Presented approach is too naïve in present form. I have several mandatory points that needs to be addressed before publication:

1)      Zhang calibration should be performed with about 20 images – not seven, why authors used 7?

We agree that 7 images might be too low. We repeated the experiment with 15 images. The new results with 15 images are included in Section 2.3. The analysis shows similar results with slightly better results for the checkerboard.

We based this number of boards on documentation of Matlab, OpenCV and Halcon which all use between 10 and 20 images. Other software like  DLR camera calibration only uses up to ten images. During capturing we made special care that all positions are unique and angles up to 30° were present to ensure good input data.  The arguments of this choice are added to the text between lines 128 and 132.

2)      How did you get the reference focal length in pixels?

We have added an explanation on how this estimation of the focal length was obtained in line 118:

A 16mm lens is used. The camera sensor has a pixel size of 5.3 µm/pixel which gives a theoretical focal length of 3018 pixels (3018 pixels = 16mm / (5.3 µm/pixel))

3)      Please provide full camera calibration followed by measurement of real reference distances. Only by measuring of known distance you can prove that your calibration is better.

We have performed an additional validation experiment in which an LCD screen was mounted on a translation stage with a position accuracy of 0.01 mm. The checkerboard is detected in 20 different positions between 200mm and 500mm from the camera. Between each pair of positions, there was a translation of 10mm. Next, the relative translation of the checkerboard is calculated with calibration parameters of the method using Gray-code and the method using checkerboards. The mean relative error on the translation with the Gray-code parameters is 0.78mm with a standard deviation of 0.32mm. The mean relative error using checkerboard parameters is 1.65mm with a standard deviation of 0.93mm. This means that the calibration parameters obtained using Gray-code are closer to the real values than the calibration parameters obtained using checkerboards.

(new paragraph added from line 171)

4)      Please discuss and provide reference data about flatness of your calibration model. According to my knowledge flatness of calibration model and accuracy of marker detection are main factors of calibration quality. In practice for precise measurement the special calibration bodies are manufactured and measured by reference method. Later knowing their exact dimension calibration is performed. How you measured planarity error of your calibration model?

Our method does not try to replace the high-quality calibration boards you refer to (additional explanation added around line 74). It does however try to improve low-cost calibration boards where users are not able to use specially calibrated bodies.

In abstract you have “whick” word.

We have corrected the typo.

Reviewer 2 Report

Authors described a novel method to calibrate a pinhole monocular camera based on Gray code pattern. The paper is clear and easy to read, nevertheless the experimental results sometimes appear to be choppy.

Other minor observation are:

At line 35 the sentence "Other publications use LCD screens in combination with circle patterns .." should be positioned before discussing how the authors overcome the current state of the art of camera calibration approaches.

At line 38 you may also consider monocular camera calibration as a central issue for real time 3D pose estimation (DOI: 10.1016/j.jvcir.2017.06.008).

At line 78 authors cite the paper [19] but it is not clear what [19] is.

Section 2 is missing

At page 6 if the third experiment (out of 7) was an outlier you should have repeated the test in order to present a statistically significant result.

Author Response

Thank you for your interest and reviewing the paper. Below you can find answers to your comments and suggestions.

Authors described a novel method to calibrate a pinhole monocular camera based on Gray code pattern. The paper is clear and easy to read, nevertheless the experimental results sometimes appear to be choppy.

In order to present the experimental results in a more systematic way, we have re-written the section. More in particular, we have divided the section into four sub-sections.

The experiments and experimental setup now also provides more information about why specific parameters are chosen (lines 127-132 / Figure 4, lines 140-150).

Also, a new experiment is executed with more calibration patterns.

Other minor observation are:

At line 35 the sentence "Other publications use LCD screens in combination with circle patterns .." should be positioned before discussing how the authors overcome the current state of the art of camera calibration approaches.

We agree that it is better to position this sentence before the state-of-the-art discussion. Therefore we repositioned it and rewrote this part. (+- line 32)

At line 38 you may also consider monocular camera calibration as a central issue for real time 3D pose estimation (DOI: 10.1016/j.jvcir.2017.06.008).

Indeed, this reference shows the importance of camera calibration for 3D pose estimation. It is added to the text.

At line 78 authors cite the paper [19] but it is not clear what [19] is.

This is a reference to the original patent describing Gray-code. The reference style is fixed and now also holds the patent number.

Section 2 is missing

Fixed, (should not have been visible in your version, thank you for pointing this out)

At page 6 if the third experiment (out of 7) was an outlier you should have repeated the test in order to present a statistically significant result.

New experiments are executed with more calibration positions (15 compared to 7) to prevent outliers. The results were similar with slightly better results for checkerboard patterns.

Reviewer 3 Report

The paper deals with new camera calibration approach. In the present form the paper present many lacks. Starting from the methdological section authors should provide more details regarging the specific steps involved in the proposed approach in order to better justify the choice of specific parameters and sollution rathern than others available, Also in the experimenatl validaion a more structured experimental setting should be used

Author Response

Thank you for your interest and reviewing the paper. Below you can find answers to your comments and suggestions.

Note: In our original submission we used 7 calibration positions in our validation. In response to another reviewer we did new experiments with 15 positions. The analysis showed similar results with slightly better results for the checkerboard.

The paper deals with new camera calibration approach. In the present form the paper present many lacks. Starting from the methodological section authors should provide more details regarding the specific steps involved in the proposed approach in order to better justify the choice of specific parameters and solution rather than others available, Also in the experimental validation a more structured experimental setting should be used

We rewrote large parts of the paper to explain better why specific parameters are chosen. Also new experiments are added to the paper. This is mainly visible in section 2 (Experiments):

       It now is now divided into 4 subsections.

       The experimental setup now also contains references and arguments why 15 camera positions are used.

       A new section is added with an experiment showing that using more points improves the stability of the calibration. This section also explains why a subset of 700000 points is used in the other experiments.

       The original table containing results of only focal lengths now also holds parameters like radial distortion parameters.

The problem statement and introduction (line 70-75) now also explains that the methodology does not try to replace high-quality industrial-grade calibration boards, but only low cost printed boards.

Round 2

Reviewer 1 Report

The authors have greatly improved their work. That's why I recommend it for publication.

Reviewer 3 Report

The scientific level of the paper has been improved